# Lipid levels after childbirth and association with number of children: A population-based cohort study

**Aleksandra Pirnat** [1] *, **Lisa A. DeRoo** [1], **Rolv Skjærven** [2], **Nils-Halvdan Morken** [3,4]

**1** Department of Global Public Health and Primary Care, University of Bergen, Bergen, Norway, **2** Centre for Fertility and Health, Norwegian Institute of Public Health, Oslo, Norway, **3** Department of Clinical Science, University of Bergen, Bergen, Norway, **4** Department of Obstetrics and Gynecology, Haukeland University Hospital, University of Bergen, Bergen, Norway

* pirnatdraleksandra@gmail.com

## Abstract

### Objective

Low parity women are at increased risk of cardiovascular mortality. Unfavourable lipid profiles have been found in one-child mothers years before they conceive. However, it remains unclear whether unfavourable lipid profiles are evident in these women also after their first birth. The aim was to estimate post-pregnancy lipid levels in one-child mothers compared to mothers with two or more children and to assess these lipid's associations with number of children.

### Methods

We used data on 32 618 parous women (4 490 one-child mothers and 28 128 women with ≥2 children) examined after first childbirth as part of Cohort of Norway (1994–2003) with linked data on reproduction and number of children from the Medical Birth Registry of Norway (1967–2008). Odds ratios (ORs) with 95% confidence intervals (CIs) for one lifetime pregnancy (vs. ≥2 pregnancies) by lipid quintiles were obtained by logistic regression and adjusted for age at examination, year of first birth, body mass index, oral contraceptive use, smoking and educational level.

### Results

Compared to women with the lowest quintiles, ORs for one lifetime pregnancy for the highest quintiles of LDL and total cholesterol were 1.30 (95%CI: 1.14–1.45) and 1.43 (95%CI: 1.27–1.61), respectively. Sensitivity analysis (women <40 years) showed no appreciable change in our results. In stratified analyses, estimates were slightly stronger in overweight/obese, physically inactive and women with self-perceived bad health.

### Conclusions

Mean lipid levels measured after childbirth in women with one child were significantly higher compared to mothers with two or more children and were associated with higher probability

---

**Data Availability Statement:** Data are available upon request due to legal and ethical restrictions imposed by Norwegian law and regional ethical committee related to patient confidentiality. Researchers who are interested in using CONOR

data for research purposes can apply for access to the CONOR steering committee at: conor@fhi.no. Guidelines for access are available at: https://www.fhi.no/globalassets/dokumenterfiler/studier/conor/guidelines-for-access-to-conor-materials.pdf.

**Funding:** This work was supported by the Norwegian Association for Public Health with doctoral scholarship 2013.ST.056 to AP. The Norwegian Association for Public Health had no role in design and conduct of the study; in collection, analysis, interpretation of data; or in the preparation, review, or approval of the manuscript.

**Competing interests:** The authors have declared that no competing interests exist.

of having only one child. These findings corroborate an association between serum lipid levels and one lifetime pregnancy (as a feature of subfecundity), emphasizing that these particular women may be a specific predetermined risk group for cardiovascular related disease and death.

## Introduction

A women's reproductive history may affect future cardiovascular disease (CVD) risk [1, 2, 3]. Studies suggest an association between subfertility and later incidence of CVD [4]. Substantial increase in CVD mortality has been found in women with only one child [2, 5, 6, 7] and lipid disorders are suggested to play a role in both subfertility and later CVD development [1, 4, 8, 9].

Animal studies have reported association between dyslipidemia and infertility, showing sterility in high-density-lipoprotein (HDL) receptor-deficient female mice [10]. Emerging research further support involvement of lipids in human fertility [11, 12, 13, 14, 15, 16]. Cholesterol is known to be essential for the process of steroidogenesis, and serum free cholesterol concentrations have been associated with fecundity in both sexes [11, 15]. HDL cholesterol is, along with Apolipoprotein b (Apo b) [17, 18], the predominant lipoprotein in ovarian follicles, and is associated with embryo quality and fertility treatment outcomes [16, 19]. Human studies have reported appreciably higher clinical pregnancy rate and number of top-quality embryos in high Apo b patients undergoing fertility treatment, compared with low Apo b patients, even after exclusion of ovarian-related disorders [17].

Lipid profile is susceptible to change during women's lifespan, influenced by pregnancy [3, 8, 20, 21] and menopause [22, 23]. Estrogen is recognized to induce an early increase of low-density-lipoprotein (LDL) receptors and enhance biliary secretion of cholesterol, with its decline in menopause leading to increased levels of both lipids [22]. There are conflicting evidence for plasma lipid changes associated with parity [3, 20, 21, 24], with most analyses using nulliparous women as the reference group. Although relevant from the aspect of total parity, this design has limited the ability of prior studies to identify the high-risk group of women having only one-child (as a feature of subfecundity). We have previously found that one-child mothers have unfavorable lipid profiles compared to women with two or more children, years before they conceive [25]. Given the effect of pregnancy on lipid levels [3, 20, 21], as well as their change during a woman's lifecycle [22], it is not clear whether unfavorable lipid profiles are evident in one-child mothers also after their first birth.

Our aim was to estimate post-pregnancy lipid levels in one-child mothers compared to mothers with two or more children and to assess these lipid's associations with number of children.

## Materials and methods

### Data sources

We used data from Cohort of Norway (CONOR) linked with the Medical Birth Registry of Norway (MBRN). CONOR is a population-based collection of health data with blood samples and lifestyle questionnaires obtained from participants aged 20 years or more, residing in different regions in Norway during 1994–2003 [26]. Women participating in the current study ≤69 years were examined after their first childbirth (singleton gestation ≥22 weeks) and

provided questionnaire data on smoking, oral contraceptive use, years of attained education (in Norway, the first 10 years are mandatory) and lifestyle factors. The health examination included standardized measurements of height, weight and non-fasting lipid levels.

All deliveries in Norway are subject to compulsory reporting to the MBRN since 1967. The registry contains information on maternal health prior to pregnancy, health and complications during pregnancy and perinatal data [27]. Registration is completed on a standardized form by the attending midwife or obstetrician. Data on in-vitro-fertilization (IVF) were available from 1988. A unique personal identification number (given to all Norwegian residents) enabled linkage of data from CONOR with the MBRN and identification of all births to each participating woman during 1967 to 2008. All included women from CONOR were followed for the occurrence of a second birth until 2008. One-child mothers were identified as women being 7 years out from their first pregnancy and with no additional births in the MBRN. In Norway >95% of women will have their second pregnancy within 7 years [5]. Given that the aim was to explore the association between post-pregnancy lipid status and number of live-born children, stillbirths and/or abortions were not included.

The study was approved by the ethical review board REK-Vest (Ref number 2013/118) and access to data was granted by the steering committee of CONOR and by the MBRN. Our study used banked blood samples collected in CONOR, and subjects were not re-contacted for the analysis. Written informed consent included use for research and linkage to health registries, and was obtained for each participant. Personal identification numbers are omitted from data when used in research purposes. The CONOR recruitment process and the obtainment of written informed consent are described in detail elsewhere [26].

## Health measurements

Non-fasting blood samples were obtained by trained personnel and analyzed on a Hitachi 911 Auto Analyzer (Hitachi, Mito; Japan) [26]. Serum concentrations of total cholesterol, HDL cholesterol and triglyceride (TG) were analyzed subsequent to sampling, with the use of reagents from Boehringer Mannheim (Mannheim, Germany). Total cholesterol and HDL cholesterol were measured by applying an enzymatic colorimetric cholesterolesterase method, with HDL cholesterol measured after precipitation with phosphortingsten and magnesium ions. An enzymatic colorimetric method was applied for measuring TG, while glucose was measured by using an enzymatic hexokinase method [28].

The day-to-day coefficients of variation were: total cholesterol: 1.3%-1.9%; HDL cholesterol: 2.4%; TG: 0.7%-1.3% and glucose: 1.3–2.0%. We calculated LDL using the Friedewald formula [29]: Total serum cholesterol minus HDL cholesterol minus one fifth of the TG concentration. LDL cholesterol levels were calculated only for participants with TG concentrations < 4.5mmol/l (due to the lower precision of calculation with highly increased TG levels) [29]. We additionally used non-HDL cholesterol levels (calculated as total cholesterol minus HDL cholesterol) as a useful toll in individuals with higher TG levels [30]. TG/HDL ratio was expressed in mmol/l.

Height and weight was measured by trained personnel with the participants wearing light clothes and no shoes; height to the nearest 1.0 cm and weight to the nearest 0.5 kg. Body mass index (BMI) was calculated as weight in kilogram/(height in meters)$^2$.

All CONOR participants signed a written informed consent for research and linkage with health registries when they participated in the survey. This study used banked blood samples collected in CONOR, and subjects were not re-contacted for this analysis. The CONOR recruitment process and the obtainment of written informed consent are described in detail elsewhere [26].

## Statistical analyses

Baseline characteristics were presented as means with standard deviations (continuous data) and numbers with percentages (categorical data). Differences between lipid quintiles were assessed by p values (Wald test) and between one-child mothers and mothers with two or more children, using Chi-square test and t-test, where appropriate.

We used logistic regression to calculate odds ratios (ORs) for one lifetime pregnancy by lipid levels. Estimates were adjusted for mother's age at examination (linear term), year of first birth (linear term), body mass index (BMI) (linear term), oral contraceptive use (now, previously, never), smoking (at examination: yes, no), education (≤11 years (low), >11 years (high)) and time since last meal (linear term). Besides accounting for time elapsed since first birth, year of first birth was also used as a proxy for generational/environmental factors [31, 32]. Oral contraceptive (OC) use was defined as current use of OC, previous use or never. Effect of BMI (<25 and ≥25), self-perceived health (good and bad) and education (high and low) were also assessed in stratified analyses. Answers 'poor' and 'not so good' were classified as 'bad', while 'good' and 'very good' were classified as 'good' perceived current health. We performed sensitivity analysis on women <40 years of age to explore the effect of menopause on women's lipid profile. Missing data were low for the majority of parameters, and were excluded from the main analyses, except for the OC use. Due to higher numbers of missing values for the glucose, this variable was excluded from further analyses.

We compared the occurrence of IVF in first pregnancy, diabetes, use of antihypertensive medications, polycystic ovary syndrome (PCOS), and thyroid disease between one-child mothers and women with two or more children. We also excluded women using antihypertensives in main analyses.

In sub-analyses we explored the impact of past year physical activity (≤1 hour per week and ≥1 hour per week) and alcohol use (≤1 time per month and >1 time per month). We also excluded women with reported hearth attack and/or angina in siblings and/or parents, with additional exclusion of women with diabetes in parents.

In order to assess how robust the associations are to potential unmeasured confounding, we calculated E-values [33] for both the adjusted main analyses and adjusted sensitivity analysis on women <40 years of age. The E-vale is defined as "the minimum strength of the association, on the risk ratio scale, that unmeasured confounder would need to have with both the exposure and the outcome to fully explain away this exposure-outcome association, conditional on the measured covariates" [32, 33].

## Results

We identified 44 126 women ≤69 years at examination and with viable singleton first births (≥22 weeks of gestation) that had participated in CONOR. After exclusion of women that were pregnant or had unknown pregnancy status, women with missing lipid assessments and women on lipid lowering drugs we had 32 618 women for our main analyses. A flow chart of inclusions and exclusions is presented in Fig 1.

One-child mothers were older at examination and had a shorter time span from first childbirth to examination, compared to women with two or more births. They had higher education but were more frequent smokers and reported more often having bad health. Mean values of all examined lipids and glucose, except TG/HDL ratio, were higher in one-child mothers (Table 1).

Adjusted ORs with 95% CIs for having one lifetime pregnancy (vs. ≥2 pregnancies) by lipid quintiles are presented in Fig 2 (numbers and crude estimates in S1 Table). The OR of one lifetime pregnancy for women with the highest LDL quintile (compared with women with the

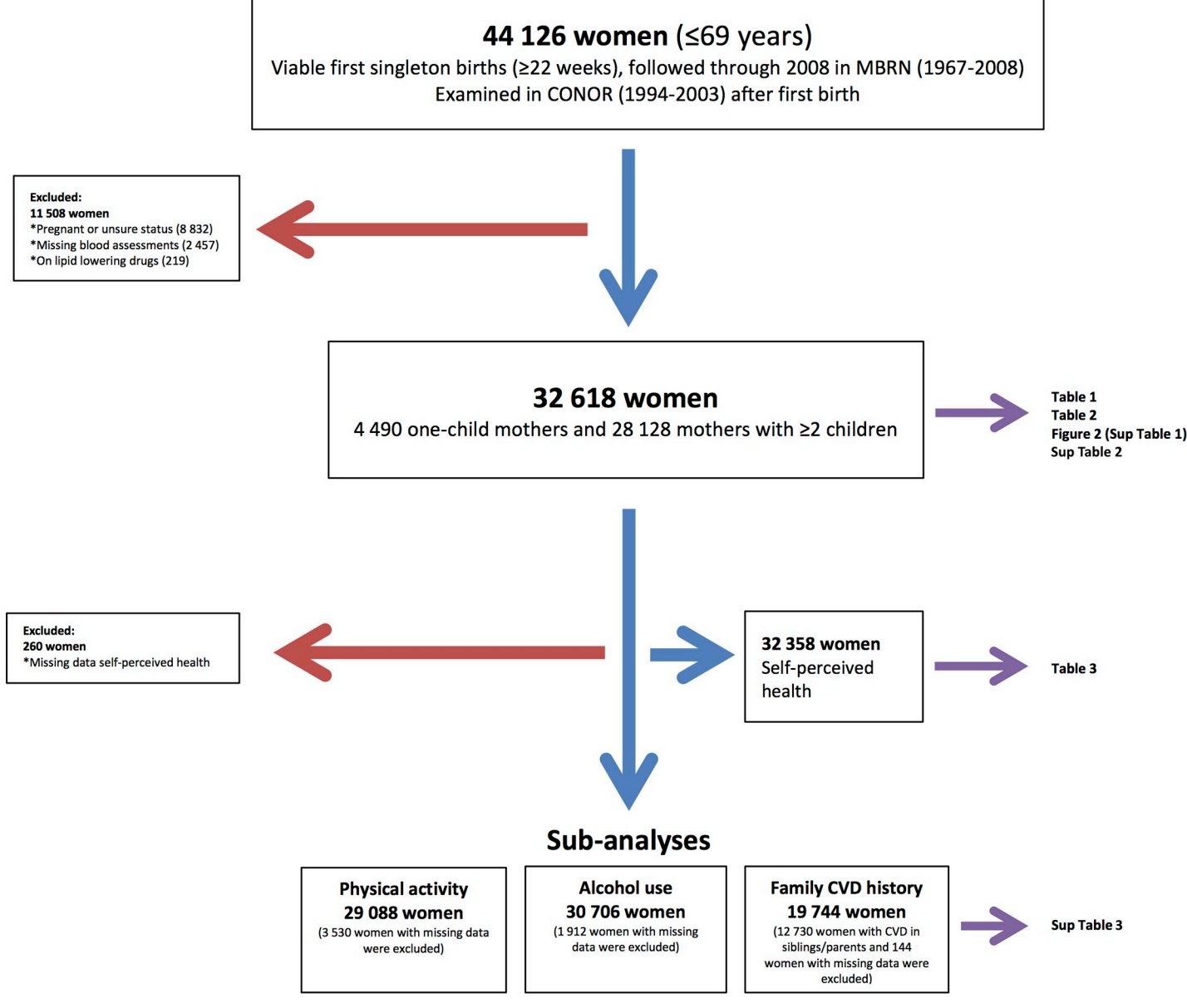

**Fig 1. Flow chart of inclusions and exclusions.**

lowest quintile) was 1.30 (95% CI 1.14–1.45), while 1.24 (95% CI 1.12–1.37) and 1.43 (95% CI 1.29–1.59) for the two highest quintiles of total cholesterol. However, there were significant differences in ORs of one lifetime pregnancy between quintiles also for HDL and TG/HDL ratio in addition to LDL and total cholesterol.

Stratified analyses by BMI at examination are presented in Table 2. Associations were strengthened for levels of LDL, total cholesterol and TG in women with BMI ≥25. ORs of one lifetime pregnancy for women with post-pregnancy lipids above clinically recommended levels of LDL and total cholesterol were: 1.32 (95% CI 1.08–1.60) and 1.46 (95% CI 1.20–1.78) for fourth and fifth quintile of LDL and 1.41 (95% CI 1.16–1.71), 1.45 (95% CI 1.20–1.76) and 1.62 (95% CI 1.33–1.97) for third to fifth quintiles of total cholesterol. For the highest quintile of

**Table 1. Characteristics of 32 618 parous Norwegian women, Cohort of Norway, 1994–2003.** Values are numbers (percentages) unless stated otherwise.

| Mean values | 4490 | 28128 | p |
|---|---|---|---|
|  | one child mothers | women with ≥ 2 children |  |
| Age (SD) at examination | 42.0 (7.1) | 40.8 (6.9) | <0.001 |
| Years (SD) from first pregnancy to examination | 14.4 (8.2) | 16.3 (7.7) | <0.001 |
| Body mass index (SD) at examination[a] | 25.1 (4.6) | 25.0 (4.0) | 0.24 |
| Oral contraceptive use |  |  |  |
| now | 318 (7.1) | 2 164 (7.7) | 0.28 |
| previously | 2 730 (60.8) | 16 973 (60.3) |  |
| never | 1 280 (28.5) | 7 834 (27.8) |  |
| missing | 162 (3.6) | 1 157 (4.1) |  |
| Smoking at examination |  |  |  |
| yes | 1 987 (44.5) | 9 415 (33.7) | <0.001 |
| now | 2 476 (55.5) | 18 510 (66.3) |  |
| missing | 27 (0.6) | 203 (0.7) |  |
| Education |  |  |  |
| <11 years (low) | 2 086 (46.4) | 13 976 (49.7) | <0.001 |
| ≥11 years (high) | 2 362 (52.6) | 13 978 (49.5) |  |
| missing | 42 (0.9) | 234 (0.8) |  |
| LDL (SD) mmol/l | 3.7 (1.0) | 3.6 (0.9) | <0.001 |
| Total cholesterol (SD) mmol/l | 5.5 (1.1) | 5.3 (1.0) | <0.001 |
| TG (SD) mmol/l | 1.3 (0.7) | 1.2 (0.7) | 0.03 |
| HDL (SD) mmol/l | 1.5 (0.4) | 1.4 (0.4) | <0.001 |
| TG/HDL (SD) mmol/l | 3.3 (1.4) | 3.4 (1.4) | 0.14 |
| Self-perceived health |  |  |  |
| god | 3 430 (76.4) | 22 788 (81.0) | <0.001 |
| bad | 1 020 (22.7) | 5 120 (18.2) |  |
| missing | 40 (0.9) | 220 (0.8) |  |
| Glucose (SD) mmol/L | 5.16 (1.1) | 5.09 (0.9) | <0.001 |
| missing | 941 (20.1) | 4 300 (15.2) |  |

[a]Missing data on 51 case of BMI.

TG OR of having one lifetime pregnancy was 1.25 (95% CI 1.03–1.53). Associations between lipid quintiles and having only one child in women with BMI<25 were only slightly attenuated from the overall results. Stratified analyses on self-perceived health are presented in Table 3. In women reporting good health, ORs of one lifetime pregnancy were similar to the main results. In women reporting bad health, ORs of one lifetime pregnancy for values above clinically recommended range of LDL, total cholesterol and TG were slightly increased. Additional analyses on non-HDL cholesterol showed similar results as for LDL levels (S2 Table). Stratification on level of education showed increased ORs among low educated women, while the higher probability of one child persisted in high-educated women, although attenuated (LDL (highest quintile): OR 1.21 (95% CI 1.02–1.43); Total cholesterol (highest quintile): OR 1.29 (95% CI 1.09–1.53)).

One-child mothers had significantly more IVF in first pregnancy (1.3% vs. 0.1%, p<0.001), were more frequent users of antihypertensive medications (3.6% vs. 2.9%, p = 0.01), had slightly higher proportion of stroke (0.6% vs. 0.4%, p = 0.05) and a significantly lower proportion of thyroid disease (0.5% vs. 1.0%, p<0.001), compared to women with two or more children. Exclusion of all women with thyroid disease from our main analyses had no effect on

a

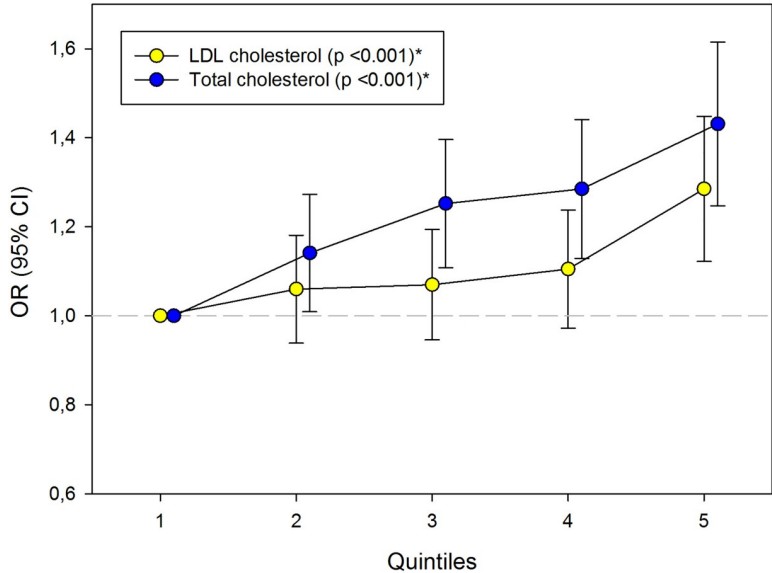

b

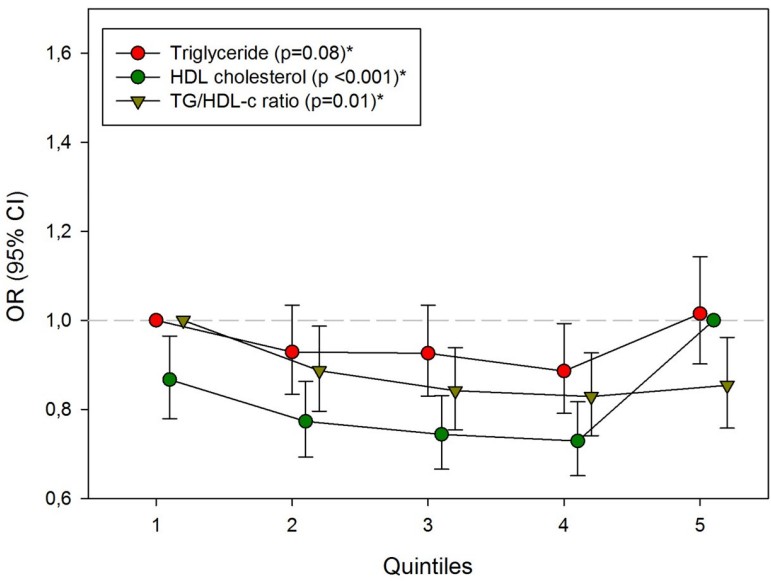

*p for difference between categories

**Fig 2. Adjusted odds ratios (ORs) with 95% confidence interval (CI) for one lifetime pregnancy by lipid quintiles in 32 618 women (≤69 years of age) examined in Cohort of Norway during 1994–2003.** All estimates were adjusted for age at examination, year of the first birth, body mass index (linear term), oral contraceptive use, smoking and educational level. a) Low-density lipoprotein (LDL) and total cholesterol, b) Triglyceride (TG), high-density lipoprotein (HDL) cholesterol and TG/HDL cholesterol ratio.

**Table 2. Adjusted odds ratio (OR) with 95% confidence interval (CI) for one lifetime pregnancy by lipid quintiles stratified by BMI (kg/m$^2$), Cohort of Norway, 1994–2003, Estimates were obtained by logistic regression and adjusted for age at examination, year of first birth, oral contraceptive use, smoking, educational level and time since last meal.**

| Lipid quintiles in mmol/l | BMI < 25 (N = 18 938) | | | | BMI ≥ 25 (N = 13 629) | | | |
|---|---|---|---|---|---|---|---|---|
| | 1 child mothers (%) | ≥ 2 children mothers | total mothers | OR (95%CI) | 1 child mothers (%) | ≥ 2 children mothers | total mothers | OR (95%CI) |
| LDL cholesterol | | | | | | | | |
| ≤ 2.87 | 674 (12.8) | 4600 | 5274 | 1.0 reference | 206 (11.3) | 1619 | 1825 | 1.0 reference |
| 2.88–3.38 | 609 (13.6) | 3856 | 4465 | 1.02 (0.90–1.16) | 316 (13.5) | 2027 | 2343 | 1.18 (0.96–1.45) |
| 3.39–3.89 | 545 (13.9) | 3369 | 3914 | 1.08 (0.95–1.24) | 368 (12.6) | 2543 | 2911 | 1.10 (0.90–1.35) |
| 3.90–4.56 | 423 (13.6) | 2686 | 3109 | 1.00 (0.87–1.16) | 453 (13.9) | 2792 | 3245 | 1.32 (1.08–1.60) |
| ≥ 4.57 | 353 (16.2) | 1823 | 2176 | 1.23 (1.05–1.45) | 537 (16.2) | 2768 | 3305 | 1.46 (1.20–1.78) |
| Total cholesterol | | | | | | | | |
| ≤ 4.60 | 608 (12.2) | 4359 | 4967 | 1.0 reference | 232 (10.9) | 1894 | 2126 | 1.0 reference |
| 4.61–5.14 | 588 (13.1) | 3895 | 4483 | 1.08 (0.95–1.23) | 317 (12.8) | 2149 | 2466 | 1.30 (1.06–1.56) |
| 5.15–5.69 | 571 (14.2) | 3453 | 4024 | 1.19 (1.04–1.36) | 412 (14.2) | 2492 | 2904 | 1.41 (1.16–1.71) |
| 5.70–6.39 | 471 (14.7) | 2722 | 3193 | 1.23 (1.06–1.42) | 424 (13.8) | 2637 | 3061 | 1.45 (1.20–1.76) |
| ≥ 6.40 | 366 (16.1) | 1905 | 2271 | 1.37 (1.17–1.61) | 495 (16.1) | 2577 | 3072 | 1.62 (1.33–1.97) |
| TG (Triglyceride) | | | | | | | | |
| ≤ 0.74 | 760 (14.4) | 4507 | 5267 | 1.0 reference | 195 (12.1) | 1409 | 1604 | 1.0 reference |
| 0.75–0.98 | 623 (13.2) | 4074 | 4697 | 0.87 (0.77–0.99) | 282 (13.2) | 1849 | 2131 | 1.12 (0.89–1.39) |
| 0.99–1.27 | 533 (13.6) | 3380 | 3913 | 0.88 (0.77–1.00) | 354 (13.6) | 2245 | 2599 | 1.10 (0.89–1.37) |
| 1.28–1.76 | 404 (13.0) | 2701 | 3105 | 0.83 (0.72–0.96) | 451 (13.6) | 2859 | 3310 | 1.07 (0.88–1.32) |
| ≥ 1.77 | 284 (14.5) | 1672 | 1956 | 0.95 (0.81–1.12) | 598 (15.0) | 3387 | 3985 | 1.25 (1.03–1.53) |
| HDL cholesterol | | | | | | | | |
| ≤ 1.19 | 317 (12.0) | 2315 | 2632 | 0.87 (0.77–0.99) | 577 (14.5) | 3397 | 3974 | 0.87 (0.71–1.07) |
| 1.20–1.38 | 465 (13.6) | 2962 | 3427 | 0.75 (0.65–0.86) | 395 (12.7) | 2713 | 3108 | 0.85 (0.70–1.03) |
| 1.39–1.55 | 478 (12.5) | 3344 | 3822 | 0.76 (0.67–0.88) | 348 (13.0) | 2314 | 2662 | 0.77 (0.64–0.93) |
| 1.56–1.79 | 594 (14.2) | 3589 | 4183 | 0.66 (0.56–0.77) | 294 (13.9) | 1810 | 2104 | 0.85 (0.71–1.01) |
| ≥ 1.80 | 750 (15.4) | 4124 | 4874 | 1.0 reference | 266 (14.9) | 1515 | 1781 | 1.0 reference |
| TG/HDL-c ratio | | | | | | | | |
| ≤ 0.45 | 806 (14.9) | 4609 | 5415 | 1.0 reference | 196 (12.5) | 1365 | 1561 | 1.0 reference |
| 0.46–0.64 | 613 (13.6) | 3875 | 4488 | 0.84 (0.75–0.96) | 282 (14.2) | 1702 | 1984 | 1.03 (0.83–1.29) |
| 0.65–0.90 | 524 (13.1) | 3463 | 3987 | 0.83 (0.73–0.95) | 327 (12.7) | 2245 | 2572 | 0.94 (0.76–1.16) |
| 0.91–1.37 | 420 (13.3) | 2735 | 3155 | 0.78 (0.68–0.90) | 468 (13.9) | 2881 | 3349 | 0.99 (0.81–1.21) |

*(Continued)*

**Table 2.** (Continued)

| Lipid quintiles in mmol/l | BMI < 25 (N = 18 938) | | | | BMI ≥ 25 (N = 13 629) | | | |
|---|---|---|---|---|---|---|---|---|
| | 1 child mothers (%) | ≥ 2 children mothers | total mothers | OR (95%CI) | 1 child mothers (%) | ≥ 2 children mothers | total mothers | OR (95%CI) |
| ≥ 1.38 | 241 (12.7) | 1652 | 1893 | 0.77 (0.65–0.91) | 607 (14.6) | 3556 | 4163 | 1.07 (0.89–1.29) |

results. Diabetes (1.1% vs. 0.9%) and history of heart attack (0.2% vs. 0.1%) were not significantly different in one-child mothers and women with two or more births. There was only one case of PCOS registered in our sample. Exclusion of women on antihypertensive therapy did not alter the main results.

**Table 3. Adjusted odds ratio (OR) with 95% confidence interval (CI) for one lifetime pregnancy by lipid quintiles, Cohort of Norway, 1994–2003.** Data stratified by self-perception of health (32 358 women), analyzed by logistic regression, adjusting for age at examination, year of first birth, body mass index (linear term), oral contraceptive use, smoking, educational level and time since last meal.

| Lipid quintiles (mmol/l) | 1 child mothers (%) | ≥ 2 children mothers | total mothers | Good health (N = 26 218) OR (95% CI) | 1 child mothers (%) | ≥ 2 children mothers | total mothers | Bad health (N = 6 140) OR (95%CI) |
|---|---|---|---|---|---|---|---|---|
| LDL cholesterol | | | | | | | | |
| ≤ 2.87 | 706 (11.8) | 5276 | 5982 | 1.0 reference | 169 (15.8) | 898 | 1067 | 1.0 reference |
| 2.88–3.38 | 745 (13.1) | 4935 | 5680 | 1.07 (0.95–1.21) | 172 (15.8) | 913 | 1085 | 0.97 (0.75–1.26) |
| 3.39–3.89 | 698 (12.6) | 4816 | 5514 | 1.04 (0.92–1.18) | 209 (16.4) | 1062 | 1271 | 1.14 (0.89–1.45) |
| 3.90–4.56 | 657 (13.2) | 4320 | 4977 | 1.08 (0.95–1.23) | 206 (15.5) | 1124 | 1330 | 1.14 (0.88–1.47) |
| ≥ 4.57 | 624 (15.3) | 3441 | 4065 | 1.25 (1.09–1.43) | 264 (19.0) | 1123 | 1387 | 1.38 (1.07–1.78) |
| Total cholesterol | | | | | | | | |
| ≤ 4.60 | 665 (11.3) | 5232 | 5897 | 1.0 reference | 169 (14.8) | 974 | 1143 | 1.0 reference |
| 4.61–5.14 | 728 (12.6) | 5057 | 5785 | 1.15 (1.02–1.30) | 169 (15.1) | 953 | 1122 | 1.10 (0.85–1.42) |
| 5.15–5.69 | 753 (13.4) | 4878 | 5631 | 1.21 (1.06–1.36) | 222 (17.7) | 1030 | 1252 | 1.43 (1.12–1.83) |
| 5.70–6.39 | 679 (13.8) | 4249 | 4928 | 1.28 (1.13–1.46) | 208 (16.2) | 1075 | 1283 | 1.32 (1.02–1.71) |
| ≥ 6.40 | 605 (15.2) | 3372 | 3977 | 1.38 (1.20–1.58) | 252 (18.8) | 1088 | 1340 | 1.61 (1.24–2.08) |
| TG (Triglyceride) | | | | | | | | |
| ≤ 0.74 | 779 (13.2) | 5109 | 5888 | 1.0 reference | 168 (17.8) | 774 | 942 | 1.0 reference |
| 0.75–0.98 | 719 (12.6) | 4970 | 5689 | 0.92 (0.82–1.04) | 178 (16.1) | 926 | 1104 | 0.88 (0.68–1.15) |
| 0.99–1.27 | 708 (13.3) | 4595 | 5303 | 0.95 (0.84–1.07) | 177 (15.1) | 995 | 1172 | 0.84 (0.65–1.09) |
| 1.28–1.76 | 628 (12.6) | 4363 | 4991 | 0.87 (0.76–0.99) | 217 (15.8) | 1152 | 1369 | 0.87 (0.67–1.13) |
| ≥ 1.77 | 596 (13.7) | 3751 | 4347 | 0.96 (0.84–1.11) | 280 (18.0) | 1273 | 1553 | 1.10 (0.85–1.41) |
| HDL cholesterol | | | | | | | | |
| ≤ 1.19 | 623 (12.5) | 4341 | 4964 | 0.87 (0.77–0.98) | 262 (16.5) | 1327 | 1589 | 0.80 (0.62–1.03) |
| 1.20–1.38 | 637 (12.4) | 4507 | 5144 | 0.76 (0.67–0.86) | 216 (16.0) | 1133 | 1349 | 0.76 (0.59–0.98) |
| 1.39–1.55 | 644 (12.2) | 4638 | 5282 | 0.73 (0.65–0.83) | 175 (15.1) | 983 | 1158 | 0.71 (0.56–0.91) |
| 1.56–1.79 | 713 (13.6) | 4521 | 5234 | 0.69 (0.61–0.79) | 169 (16.7) | 845 | 1014 | 0.76 (0.60–0.97) |
| ≥ 1.80 | 813 (14.5) | 4781 | 5594 | 1.0 reference | 198 (19.2) | 832 | 1030 | 1.0 reference |
| TG/HDL-c ratio | | | | | | | | |
| ≤ 0.45 | 820 (13.7) | 5169 | 5989 | 1.0 reference | 176 (18.7) | 767 | 943 | 1.0 reference |
| 0.46–0.64 | 711 (13.1) | 4707 | 5418 | 0.88 (0.78–0.99) | 174 (17.0) | 849 | 1023 | 0.81 (0.63–1.05) |
| 0.65–0.90 | 669 (12.5) | 4660 | 5329 | 0.84 (0.75–0.95) | 180 (14.9) | 1021 | 1201 | 0.78 (0.61–1.01) |
| 0.91–1.37 | 671 (13.2) | 4398 | 5069 | 0.83 (0.74–0.95) | 210 (15.2) | 1175 | 1385 | 0.72 (0.56–0.93) |
| ≥ 1.38 | 559 (12.7) | 3854 | 4413 | 0.79 (0.69–0.91) | 280 (17.6) | 1308 | 1588 | 0.93 (0.72–1.20) |

The calculated E-values for the significant estimates were as follows: main analyses—for the ORs of one lifetime pregnancy by highest quintiles of LDL and total cholesterol levels: 1.54 and 1.68, respectively; sensitivity analyses (women <40 years of age)—for the ORs of one lifetime pregnancy by highest quintiles of LDL and total cholesterol levels: 1.50 and 1.60, respectively. E-value calculations showed that an unmeasured confounder would need to have nearly four times as large an effect as maternal age (covariate with the strongest effect in the adjusted model, with Exp (B) = 1.13), and be associated with both the exposure and the outcome to completely explain away the observed associations [33].

After excluding 12 730 women with reported CVD in parents or siblings and 144 women with missing information (S3 Table), probability of one lifetime pregnancy by lipid quintiles showed almost no alteration across LDL and total cholesterol levels, with slightly stronger effect on TG. Additional exclusion of diabetes in parents had no effect on results. Stratified analyses on alcohol use showed slight modifiable effect of alcohol on lipid levels. ORs of one lifetime pregnancy for LDL (highest quintile vs lowest) in low frequent users was 1.42 (95% CI 1.20–1.69) and, 1.17 (95% CI 0.98–1.39) for high frequent users. Similar results for the highest total cholesterol quintiles were 1.55 (95% CI 1.30–1.84) and 1.34 (95% CI 1.12–1.59), respectively. In women being less physically active, OR of one lifetime pregnancy was 1.40 (95% CI 1.19–1.63) for the highest LDL quintile versus lowest and 1.58 (95% CI 1.35–1.85) for total cholesterol. In women with high physical activity similar estimates for LDL and total cholesterol were 1.14 (95% CI 0.92–1.41) and 1.27 (95% CI 1.02–1.58). Other lipids showed no substantial changes in sub-analyses.

## Discussion

Mean lipid levels measured after childbirth in women with one child were significantly higher compared to mothers with two or more children. Women with LDL cholesterol greater than 4.57 mmol/l (highest quintile) and total cholesterol level greater than 5.70 mmol/l (two highest quintiles), measured more than a decade after first childbirth, had higher probability of having only one child compared to women with the lowest quintile levels. Supportive of studies that suggest the role of lipids in human fertility [8, 9, 10, 11, 12, 13, 14], these findings potentiate the dose-response lipid effect, implicating potentially negative fertility impact of clinically abnormal levels of lipids.

The increased probability for being one-child mother in women with the highest LDL quintiles, years after childbirth, is consistent with our previous findings of elevated LDL in one-child mothers examined prior to conception [25]. The increased OR for the highest total cholesterol levels, however, contrasts our previous findings. This could be due to different roles and levels of cholesterol during different stages of a woman's reproductive life, as well as decreasing estrogen levels while approaching menopause [22, 34]. Estrogen deprivation in menopause may lead to increased total and LDL levels [22], and we examined the menopausal effect in a sensitivity analysis, including only women < 40 years of age. We found that the results were only slightly attenuated from our main results (LDL (highest quintile): OR 1.23 (95% CI 0.98–1.54), total cholesterol (highest quintile): OR 1.36 (95% CI (1.09–1.70)), suggesting that menopause is not the major driver of the observed associations. Aligned with this, recent examination of the association between pregnancy and life course lipid trajectories reported no meaningful change of the results when accounted for menopausal transition [20]. Our results of increased cholesterol levels are in line with previous reports from the LIFE study [8] of higher proportion of women with menstrual irregularities in the highest quartiles of free cholesterol, as well as the association of hypercholesterolemia with ovarian infertility [35]. Some previous studies have reported no consistent association between parity and LDL/TG

levels [3, 36], while others, with longer follow-up, have found an association between declining total cholesterol levels by parity [36] and associations between primiparity and levels of total cholesterol and LDL [21]. Although unfavorable glucose levels in our study among one-child mothers is not uncommonly seen finding in dyslipidemias, caution is needed in interpretation of this result due to high number of missing data. We found no effect on probability of one lifetime pregnancy across HDL and TG/HDL levels. This is consistent with a decreasing and still unclear effect of higher parity on HDL levels [2, 20, 21, 24]. Although age–related factors are suggested to play a role in the change of HDL fractions in follicular fluid [18], several studies have reported the highest magnitude of the HDL drop associated with first birth, independent of maternal age [20, 21, 24]. While HDL concentrations in follicular fluid have been found to correlate with plasma levels [17], exactly how HDL content is influenced by pregnancy or may influence fertility potential is still unclear [18], and remains to be explored.

Possible mechanisms could be genetic differences or incipient dyslipidemias, which may induce excessive alterations in levels of lipoproteins associated with pregnancy [1, 3, 21]. It is suggested that the most prominent lipid changes occur following first birth [20], and that one-child mothers begin their reproductive career with unfavorable lipid profiles years before conception [25]. Progesterone during pregnancy may act to reset lipostat in the hypothalamus [37] and the placenta may convey an active role on maternal lipoprotein metabolism through fetal polymorphisms (inherited from the father) [38]. It is possible, that in some women with preexisting dyslipidemia, placental influence (expressed from paternal inherited allele) will either partly compensate for or exaggerate maternal lipid profile [38]. Hormonal changes accompanying pregnancy, related fat retention and/or redistribution and lifestyle/behavioral practices may introduce long-term changes in lipid metabolism [1, 3, 21], particularly in predisposed women.

The unfavorable metabolic milieu of obesity may also contribute to reduced fertility, decreasing probability of conception and influencing lipid profile [9, 39]. Aligned with this, our stratified results for BMI≥25 showed adverse effect of obesity on lipid levels [39]. Only slight attenuation of ORs in normal weight women with the highest LDL and total cholesterol levels supports our previous results in one child mothers, where unfavorable pre-pregnancy lipid levels were found to be associated with one lifetime pregnancy also in lean women (BMI<25) [25]. A non-manifest/genetic predisposition may be exaggerated by obesity, leading to clinically high levels of certain lipids, particularly LDL and TG. This could act through chronic low-grade inflammation, one of the hallmarks of obesity that also generates increased conditions of oxidative stress, both of which are associated with lipid modifications [40]. This is in line with studies showing that genetic risk for dyslipidemia is significantly modified by obesity [41].

Self-perceived health status is considered a strong predictor of circulatory diseases and mortality and may convey additional knowledge that is not captured by available clinical measurements [42]. Indirectly, it may also provide additional insights about possible psychosocial factors, given that women with unfavorable psychosocial status are less likely to rate their health as good [42]. Higher probability of having one lifetime pregnancy only slightly decreased compared to our main results in women who perceived their health as good. This suggests that a self-rated health factor is not determining for this association, and might be another indicator of underlying biological predisposition.

PCOS has also been linked to dyslipidemia; however, we found only one case in our study sample. The Coronary Artery Risk Development in Young Adults Study (CARDIA) suggests that lower concentrations of serum dehydroepiandrosterone sulfate (DHEAS) and dehydroepiandrosterone (DHEA) are associated with a first pregnancy rather than parity per se [3, 21]. Although increased androgen levels are seen in women in PCOS, a recent study reported

androgen-related ovulatory dysfunction in otherwise apparently healthy, eumenorrheic women, supportive of non-manifest subfertile type [43].

Exclusion of women with family history of CVD showed little effect on lipid associations, apart from a slightly stronger effect on TG (S3 Table). Mounting evidence suggests that hyper-triglyceridemia is an independent risk factor of CVD, even with well-controlled LDL levels [44]. Sub-analyses on physical activity are consistent with research suggesting modifiable effect of physical activity on lipid status [45]. Alcohol use showed stronger effect on LDL levels, while for the total cholesterol levels we found OR alteration in low frequency users and decreased OR for high frequency users. This may reflect reluctance to report drinking frequency in the low frequency group or that abstinence from alcohol is a marker of other unmeasured risk [46].

A woman's risk of developing chronic conditions increases at menopause, which may reflect cumulative impact of earlier alterations in CVD risk factors, accelerated by perimenopausal transition [34]. Increase in CVD risk in postmenopausal women is suggested to be due to increased LDL and total cholesterol levels, along with arterial remodeling and other factors [22, 47, 48]. The significantly higher mean values in nearly all the observed lipids in one-child mothers compared to mothers with two or more children indicate that worsened lipid profile among women approaching midlife is additionally exaggerated in one-child mothers. A baseline difference of only 0.41mmol/l in serum cholesterol is independently associated with a 21% excess risk of death from coronary heart disease [22, 47].

We used a large population-based cohort sample. Linked data from the MBRN provided complete registration of total reproduction and enabled identification of all births to each woman. A limitation is blood sampling in non-fasting state. However, adjusting our analyses for time since last meal showed no substantial change in results, suggesting that non-fasting lipids are not likely to introduce systematic bias. Non-fasting lipid levels are successfully used in lipid and CVD research [8, 45, 49] with non-fasting TG levels being strongly associated with incident CVD events [50]. Similarity in results obtained for non-HDL and LDL cholesterol further strengthens the role of lipids and supports the optimal performance of LDL calculations in our study (by Friedwald formula). We lacked data on C-reactive protein, apolipoprotein E genotype, and thyroid tests/antibodies, factors that may affect lipid status and fertility. However, exclusion of women with thyroid disease did not influence our results. Assessments of duration of oral contraceptive use, sex hormone status, dietary intake or stress were also not available. We had only one case of PCOS in our sample; hence, underreporting may be present. As in all observational studies, unmeasured confounding in our study cannot be excluded. However, calculated E-values indicated that any unmeasured factor would need to have nearly four times as large an effect as maternal age, and be associated with both the lipid levels and fecundity to completely explain away the observed associations [33]. Additionally, persistent higher ORs in our stratified results for both the strata of women who rate their health as good and those highly educated suggests that women's self-perceived health and education/socio-economic status are not the major determinants of the observed association in our study.

Our findings corroborate an association between serum lipid levels and one lifetime pregnancy (as a feature of subfecundity), emphasizing that these particular women may be a specific predetermined risk group for cardiovascular related disease and death [5].

## Supporting information

**S1 Table. Crude and adjusted odds ratio (OR) with 95% confidence interval (CI) for one lifetime pregnancy by lipid quintiles in 32 618 parous Norwegian women (≤69 years of age), Cohort of Norway, 1994–2003.** Estimates were obtained by logistic regression and

adjusted for age at examination, year of first birth, body mass index (linear term), oral contraceptive use, smoking, educational level and time since last meal.
(PDF)

**S2 Table. Adjusted odds ratio (OR) with 95% confidence interval (CI) for one lifetime pregnancy by non-HDL cholesterol quintiles in 32 618 parous Norwegian women ($\leq$69 years of age), Cohort of Norway, 1994–2003.** Estimates were obtained by logistic regression and adjusted for age at examination, year of first birth, body mass index (linear term), oral contraceptive use, smoking, educational level and time since last meal.
(PDF)

**S3 Table. Adjusted odds ratios (ORs) with 95% confidence interval (CI) for one lifetime pregnancy by lipid quintiles in 19 744 parous Norwegian women without reported cardiovascular disease in parents or siblings, Cohort of Norway, 1994–2003.** Estimates were obtained by logistic regression and adjusted for age at examination, year of first birth, body mass index (linear term), oral contraceptive use, smoking, educational level and time since last meal.
(PDF)

## Acknowledgments

We are thankful to the CONOR steering group who approved the study and kindly provided data for the analyses. We also thank all the women who have participated in the CONOR cohort, as well as collaborating staff who have supported data collection.

## Author Contributions

**Conceptualization:** Aleksandra Pirnat, Lisa A. DeRoo, Rolv Skjærven, Nils-Halvdan Morken.

**Formal analysis:** Aleksandra Pirnat.

**Funding acquisition:** Nils-Halvdan Morken.

**Methodology:** Aleksandra Pirnat, Lisa A. DeRoo, Rolv Skjærven, Nils-Halvdan Morken.

**Supervision:** Rolv Skjærven, Nils-Halvdan Morken.

**Writing – original draft:** Aleksandra Pirnat.

**Writing – review & editing:** Lisa A. DeRoo, Rolv Skjærven, Nils-Halvdan Morken.

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
