## [Decision Letter · Decision Letter 0]

10 Jun 2019

PONE-D-19-14622

Lipid levels after childbirth and association with number of children: a population-based cohort study

PLOS ONE

Dear Dr Pirnat,

Thank you for submitting your manuscript to PLOS ONE. After careful consideration, we feel that it has merit but does not fully meet PLOS ONE’s publication criteria as it currently stands. Therefore, we invite you to submit a revised version of the manuscript that addresses the points raised during the review process.

Please address the reviewers' very helpful comments

We would appreciate receiving your revised manuscript by Jul 25 2019 11:59PM. To enhance the reproducibility of your results, we recommend that if applicable you deposit your laboratory protocols in protocols.io, where a protocol can be assigned its own identifier (DOI) such that it can be cited independently in the future. For instructions see: http://journals.plos.org/plosone/s/submission-guidelines#loc-laboratory-protocols

We look forward to receiving your revised manuscript.

Kind regards,

Catherine Mary Schooling

Academic Editor

PLOS ONE

Journal Requirements:

2. In ethics statement in the manuscript and in the online submission form, please provide additional information about the patient records used in your retrospective study. Specifically, please ensure that you have discussed whether all data were fully anonymized before you accessed them and/or whether the IRB or ethics committee waived the requirement for informed consent. If patients provided informed written consent to have data from their medical records used in research, please include this information.

Reviewers' comments:

Reviewer's Responses to Questions

**Comments to the Author**

1. Is the manuscript technically sound, and do the data support the conclusions?

Reviewer #1: Yes

Reviewer #2: Partly

2. Has the statistical analysis been performed appropriately and rigorously? 

Reviewer #1: Yes

Reviewer #2: No

3. Have the authors made all data underlying the findings in their manuscript fully available?

Reviewer #1: Yes

Reviewer #2: No

4. Is the manuscript presented in an intelligible fashion and written in standard English?

Reviewer #1: Yes

Reviewer #2: Yes

5. Review Comments to the Author

Reviewer #1: Lipid levels after childbirth and association with number of children: a population-based

cohort study

This is a well written paper, with relatively original data. I think the paper in revised form is even more suitable for publication. I would suggest to adapt the paper, enabling to interpret it’s findings on a more pathophysiological basis, something that is now lacking and could be viewed as a caveat. Additionally I would personally present the data sometimes a bit differently, however this is largely a matter of personal choice.

The most important finding is that serum lipid levels are associated with reduced lifetime pregnancy.

Abstract:

1) Only after careful reading I realized the meaning of one lifetime pregnancy, i.e that it is “unfavorable”, since there was no clear contrast especially in the methods, results and conclusion, to what one lifetime pregnancy was compared to.

Introduction and results

2) In the introduction, line 66 to 69, there is a link with fertility, HDL and APOb, however, this crucial bit of information is not further used in the paper, especially in the discussion. It would be nice to expand the HDL findings and it’s relation with fertility further. This also relates to the findings presented in table 2 and 3. First of all I think readability of these tables could further improve by providing the reader with column percentages, thus highlighting the dose response relationship between lipid levels and lifetime pregnancy.

Next I would show the relation between HDL and lifetime pregnancy slightly different, by choosing the lowest level of FHL cholesterol as reference category rather than highest HDL levels.

3) Additionally, the reason for stratification for high vs low BMI and for good vs bad perceived health should be elucidated further. And an analysis on the overall effect of lipid levels on lifetime pregnancy should be provided.

Results and discussion.

4) The authors suggest in the discussion that a genetic predisposition line 310 may partly explain these findings. I think this part is very important. I hope the authors can provide additional analysis on the effect of the APOE gene in this beautiful cohort study on lifetime pregnancy, this would greatly improve the paper.

5) Finally the authors should provide information why there was only one patient in their cohort with PCOS (moreover it would be nice for the reader). And they should provide information on statin use and it’s effect om their findings.

Reviewer #2: In this study, the authors aimed to estimate post-pregnancy lipid levels in one-child mothers compared to mothers with two or more children and to assess these lipid’s associations with number of children.

It is an interesting study, however, I have some concerns concerning the clarity and validity of the findings.

1) It is unclear how these women were selected. Did the authors select all women with history of delivery? How about the women with history of abortion or stillbirth? If they are not included, will this affect the findings? The authors need to make it more clear and discuss more on this point.

2) In the objective, the authors seem to aim to examine bi-directional associations of number of children with lipid profile. However, in the results, they did not show the adjusted association of number of children with lipids. The main findings seem to focus on examining the other direction of association.

3) Some potential confounders, such as socioeconomic position and sex hormone status, which may affect both lipids and number of children, are not included in the model.

4) The authors mentioned E-values in the method, however, it is unclear how it is calculated. Please add more details on the calculation and result interpretation.

5) PCOS is a quite common disorder with prevalence over 4%. In this study only one PCOS case was identified. Is it possible some people did not report? If so, how will this affect the findings?

6. PLOS authors have the option to publish the peer review history of their article (what does this mean?). If published, this will include your full peer review and any attached files.

Reviewer #1: Yes: Eric van Exel

Reviewer #2: No

---

## [Author Response · Author response to Decision Letter 0]

25 Jul 2019

To the Editor-in-chief, 

Plos One

 July 2019

Dear Madam, 

Attached is a revised version of the paper entitled: “Lipid levels after childbirth and association with number of children: a population-based cohort study” (PONE-D-19-14622), which we hope will be considered for publication in Plos One.

We thank the editors and reviewers for their constructive comments and suggestions. Below is a point by point response to the posed questions and comments. 

Editor’s comments

Please include the following items when submitting your revised manuscript: A marked-up copy of your manuscript that highlights changes made to the original version. This file should be uploaded as separate file and labeled 'Revised Manuscript with Track Changes'.

Response: All changes in the manuscript are marked with red font colour and a marked-up copy has been uploaded.

Reviewer #1

This is a well written paper, with relatively original data.

Response: Thank you.

I think the paper in revised form is even more suitable for publication. I would suggest to adapt the paper, enabling to interpret it’s findings on a more pathophysiological basis, something that is now lacking and could be viewed as a caveat. Additionally I would personally present the data sometimes a bit differently, however this is largely a matter of personal choice.

The most important finding is that serum lipid levels are associated with reduced lifetime pregnancy.

Abstract:

1) Only after careful reading I realized the meaning of one lifetime pregnancy, i.e that it is “unfavorable”, since there was no clear contrast especially in the methods, results and conclusion, to what one lifetime pregnancy was compared to.

Response: Thank you for noticing this. We have now changed the appropriate sections of the abstract according to your comments. (Please see page 2).

Introduction and results

2) In the introduction, line 66 to 69, there is a link with fertility, HDL and APOb, however, this crucial bit of information is not further used in the paper, especially in the discussion. It would be nice to expand the HDL findings and it’s relation with fertility further. 

Response: We agree with the reviewer that it would be beneficial to expand the findings on HDL and the role of ApoB in relation to fertility. We have now added a few more sentences on the possible effect of these lipid fractions on female fertility in the appropriate sections; however, too extensive elaboration on ApoB and HDL is largely limited due to their still unclear role in follicular fluid and related fertility impairment (1, 2) (Please, see page 3, lines: 66-67 and page 16, lines: 311-316).

This also relates to the findings presented in table 2 and 3. First of all I think readability of these tables could further improve by providing the reader with column percentages, thus highlighting the dose response relationship between lipid levels and lifetime pregnancy.

Response: Thank you for this comment. We agree with the reviewer that this would improve readability, and have now added the percentages within the columns of one-child mothers in the appropriate tables (please, see Tables 2 and 3).

Next I would show the relation between HDL and lifetime pregnancy slightly different, by choosing the lowest level of FHL cholesterol as reference category rather than highest HDL levels. 

Response: We understand the reasoning behind this suggestion, and we find this also a valid approach (assuming that FHL abbreviation refers to HDL cholesterol?) However, given our previous study findings (on pre-pregnancy sample of women), where we show the association between pre-pregnancy HDL levels and one lifetime pregnancy (using highest HDL levels as a reference category) (3), we believe that keeping consistency in this approach would in a better way present changing patterns within different lipids-parity associations from the pre- to post-pregnancy period. 

3) Additionally, the reason for stratification for high vs low BMI and for good vs bad perceived health should be elucidated further. And an analysis on the overall effect of lipid levels on lifetime pregnancy should be provided.

Response: We agree that this would be beneficial for the reader, and have now added more details on the reasons behind these stratifications (please, see page 17, lines 329-330 and 340-343). 

As for the suggested analysis on the overall effect of lipid levels on lifetime pregnancy we unfortunately have to admit that we are not sure what the reviewer means here. We do however believe that this might have been presented in the provided Supplemental material (please see Table S1).

Results and discussion.

4) The authors suggest in the discussion that a genetic predisposition line 310 may partly explain these findings. I think this part is very important. I hope the authors can provide additional analysis on the effect of the APOE gene in this beautiful cohort study on lifetime pregnancy, this would greatly improve the paper.

Response: We agree that additional analysis on the ApoE gene effect would add value to the paper, however, the cohort did unfortunately not have data on ApoE genotype, which limited our ability to explore this effect. We have stated this in the Discussion section, under limitations (please, see page 19, line 384).

5) Finally the authors should provide information why there was only one patient in their cohort with PCOS (moreover it would be nice for the reader). And they should provide information on statin use and it’s effect om their findings.

Response: Thank you for noticing this. We agree with the reviewer that the low number of PCOS cases should be commented, and that underreporting may be present. We realize that this could be seen as a limitation, and have now added an explicit statement in the appropriate section (please, see page 19, line 388). Also, please see our response to question number 5 from Reviewer #2, below.

Regarding the information on statin use, all women on lipid lowering drugs (219 in total) were excluded from our analyses (please see flow chart (Figure 1)). 

Reviewer #2

In this study, the authors aimed to estimate post-pregnancy lipid levels in one-child mothers compared to mothers with two or more children and to assess these lipid’s associations with number of children.

It is an interesting study, however, I have some concerns concerning the clarity and validity of the findings.

1) It is unclear how these women were selected. Did the authors select all women with history of delivery? How about the women with history of abortion or stillbirth? If they are not included, will this affect the findings? The authors need to make it more clear and discuss more on this point.

Response: Thank you for this valid comment. As shown in the Figure 1, we included all women with viable singleton births ≥ 22 week of gestation, based on linked information from the MBRN. Given that the aim of this study was to asses women’s post-pregnancy lipid associations with number of liveborn children, abortions and/or stillbirths were not included. We, however, agree with the reviewer that this needs more explicit clarification, which we have now added under the Methods section (Please see page 5, lines 115-117). We also realize that pregnancy history may influence women’s future fertility patterns, however, this was beyond the scope of the present study, and was separately explored in another study of ours (4). 

 2) In the objective, the authors seem to aim to examine bi-directional associations of number of children with lipid profile. However, in the results, they did not show the adjusted association of number of children with lipids. The main findings seem to focus on examining the other direction of association. 

Response: This question is unclear to us; however, the adjusted associations are presented in detail in Supplemental Table S1, while Figure 2 is a synopsis of data from Table S1. 

 3) Some potential confounders, such as socioeconomic position and sex hormone status, which may affect both lipids and number of children, are not included in the model.

Response: We agree with the reviewer that important confounders should be taken into account. We realize that sex hormone fluctuations can be seen as such, and have now added this in our limitation section (please see page 19, line 387). As for socio-economic status, educational level is a commonly used measure of socioeconomic status in epidemiologic research (5), and can therefore serve as a proxy for socioeconomic and lifestyle factors (5). We further tried to explore this factor in stratified results, where persistent higher ORs for both the strata of women who rate their health as good and those highly educated suggests that women’s self-perceived health and education/socioeconomic status are not the major determinants of the observed association in our study. Residual confounding is a common problem in observational studies and cannot be excluded, however, by calculating E-value we tried to provide a measure of robustness of the reported association to this factor (6) (please, see also our response below). Calculated E-values indicated that any unmeasured factor would need to have nearly four times as large an effect as maternal age, and be associated with both the lipid levels and fecundity to completely explain away the observed associations. Although a strong unmeasured confounding factor could explain the association, a degree of confounding this strong seems not likely plausible, given our results from various stratified analyses. All things considered, we would argue that our findings are valid and reliable, but clearly have the outlined limitations already discussed under strengths and limitations.

4) The authors mentioned E-values in the method, however, it is unclear how it is calculated. Please add more details on the calculation and result interpretation..

Response: Calculations of the E-value was performed according to the formula by VanderWeele TJ et al. (6), who encourage the use of the E-value as a useful measure of robustness of the reported associations: “The E-value is defined as the minimum strength of association, on the risk ratio scale, that an unmeasured confounder would need to have with both the treatment and the outcome to fully explain away a specific treatment–outcome association, conditional on the measured covariates” (6). Accordingly, result interpretation in our study was performed in line with these recommendations (please, see page 14, lines 254-261). 

Calculations (for the common outcomes - >15% prevalence):

E-value=〖RR〗^(* ) "+ sqrt" {〖RR〗^(*" " )×(〖RR〗^(* " " )-1)}

* When the outcome is common (more than 15%), an approximate E-value can be obtained by replacing the risk ratio with the square root of the odds ratio, i.e. RR*≈sqrt(OR), in the E-value formula (6). 

5) PCOS is a quite common disorder with prevalence over 4%. In this study only one PCOS case was identified. Is it possible some people did not report? If so, how will this affect the findings?

Response: We agree that underreporting may be present – please also see our response to question number 5 from Reviewer #1. As for the possible effect of this on our findings, one may speculate that underreporting of PCOS in our cohort might result in overestimation of the observed effect, however it remains highly questionable whether this would substantially influence our results with the estimated 6% prevalence in the total sample (7). Extensive elaboration on this is further complicated by discrepancies in PCOS reporting, in part due to the use of various definitions of the syndrome and its subphenotypes, as well as differences between study cohorts and ethnicities (7). 

Data are available upon request due to legal and ethical restrictions imposed by Norwegian law and regional ethic committee related to patient confidentiality. Researchers who are interested in using CONOR data for research purposes can apply for access to the CONOR steering committee at: conor@fhi.no

Guidelines for access are available at: https://www.fhi.no/globalassets/dokumenterfiler/studier/conor/guidelines-for-access-to- conor-materials.pdf

All authors have fulfilled the conditions of authorship outlined in the instructions to authors. The final version of the manuscript has been approved by all the authors and there are no conflicts of interest among them. This study has not been published previously in any form, and it is not under consideration in any other journal. There are no conflicts of interest, including specific financial interests and relationships and affiliations relevant to the subject of this manuscript.

Aleksandra Pirnat, Lisa DeRoo, Rolv Skjærven, Nils-Halvdan Morken

The corresponding author is: Aleksandra Pirnat M.D., Department of Global Public Health and Primary Care, University of Bergen, 5018 Bergen, Norway

Phone : + 47 938 24 889, e-mail: pirnatdraleksandra@gmail.com

References:

 1. Von Wald T, Monisova Y, Hacker MR, Yoo SW, Penzias AS, Reindollar RR, et al. Age-related variations in follicular apolipoproteins may influence human oocyte maturation and fertility potential. Fertil Steril. 2010;93: 2354-2361.

 2. Gautier T, Becker S, Drouineaud V, Ménétrier F, Sagot P, Nofer JR, et al. Human luteinized granulosa cells secrete apoB100-containing lipoproteins. J Lipid Res. 2010;51: 2245-2252.

 3. Pirnat A, DeRoo L, Skjaerven R, Morken NH. Women’s pre-pregnancy lipid levels and number of children. BMJ Open. 2018;8: e021188.

 4. Pirnat A, DeRoo L, Skjaerven R, Morken NH. Risk of having one lifetime pregnancy and modification by outcome of pregnancy and perinatal loss. Acta Obstet Gynecol Scand. 2019;98(6):753-760.

 5. Oakes JM, Kaufman JS. Methods in social epidemiology. 1st ed. Ed. San Francisko, CA: Jossey-Bass; 2006.

 6. VanderWeele TJ, Ding P. Sensitivity Analysis in Observational Research: Introducing the E-Value. Ann Intern Med.2017;167: 268–274.

 7. Bozdag G, Mumusoglu S, Zengin D, Karabulut E, Yildiz BO. The prevalence and phenotypic features of polycystic ovary syndrome: a systematic review and meta-analysis. Hum Reprod. 2016 Dec;31(12):2841-2855.

---

## [Decision Letter · Decision Letter 1]

11 Sep 2019

[EXSCINDED]

PONE-D-19-14622R1

Lipid levels after childbirth and association with number of children: a population-based cohort study

PLOS ONE

Dear Dr Pirnat,

Thank you for submitting your manuscript to PLOS ONE. After careful consideration, we feel that it has merit but does not fully meet PLOS ONE’s publication criteria as it currently stands. Therefore, we invite you to submit a revised version of the manuscript that addresses the points raised during the review process.

Please amend the abstract as suggested 

We would appreciate receiving your revised manuscript by Oct 26 2019 11:59PM. To enhance the reproducibility of your results, we recommend that if applicable you deposit your laboratory protocols in protocols.io, where a protocol can be assigned its own identifier (DOI) such that it can be cited independently in the future. For instructions see: http://journals.plos.org/plosone/s/submission-guidelines#loc-laboratory-protocols

We look forward to receiving your revised manuscript.

Kind regards,

C Mary Schooling

Academic Editor

PLOS ONE

Reviewers' comments:

Reviewer's Responses to Questions

**Comments to the Author**

1. If the authors have adequately addressed your comments raised in a previous round of review and you feel that this manuscript is now acceptable for publication, you may indicate that here to bypass the “Comments to the Author” section, enter your conflict of interest statement in the “Confidential to Editor” section, and submit your "Accept" recommendation.

Reviewer #2: All comments have been addressed

2. Is the manuscript technically sound, and do the data support the conclusions?

Reviewer #2: Yes

3. Has the statistical analysis been performed appropriately and rigorously? 

Reviewer #2: Yes

4. Have the authors made all data underlying the findings in their manuscript fully available?

Reviewer #2: Yes

5. Is the manuscript presented in an intelligible fashion and written in standard English?

Reviewer #2: Yes

6. Review Comments to the Author

Reviewer #2: The authors have addressed my comments. Only one minor point, the description in the results of abstract "Compared to women with ≥2 pregnancies, ORs for one lifetime pregnancy for the highest quintiles of LDL and total cholesterol were 1.30 (95%CI: 1.14-1.45) and 1.43 (95%CI: 1.27-1.61), respectively." is not quite clear, I think the authors actually compared women with highest versus lowest quintiles of LDL and total cholesterol, so it should be "Compared to women with lowest quintiles of LDL and total cholesterol"rather than "compared to women with ≥2 pregnancies".

7. PLOS authors have the option to publish the peer review history of their article (what does this mean?). If published, this will include your full peer review and any attached files.

Reviewer #2: No

---

## [Author Response · Author response to Decision Letter 1]

22 Sep 2019

To the Editor-in-chief, 

Plos One

 September 2019

Dear Madam, 

Attached is a revised version of the paper entitled: “Lipid levels after childbirth and association with number of children: a population-based cohort study” (PONE-D-19-14622R1), which we hope will be considered for publication in Plos One.

We thank the editors and reviewers for their constructive comments and suggestions. Below is a point by point response to the posed questions and comments. 

Editor’s comments

Please include the following items when submitting your revised manuscript: A marked-up copy of your manuscript that highlights changes made to the original version. This file should be uploaded as separate file and labeled 'Revised Manuscript with Track Changes'.

Response: All changes in the manuscript are marked with red font colour and a marked-up copy has been uploaded.

Reviewer # 2

The authors have addressed my comments. Only one minor point, the description in the results of abstract "Compared to women with ≥2 pregnancies, ORs for one lifetime pregnancy for the highest quintiles of LDL and total cholesterol were 1.30 (95%CI: 1.14-1.45) and 1.43 (95%CI: 1.27-1.61), respectively." is not quite clear, I think the authors actually compared women with highest versus lowest quintiles of LDL and total cholesterol, so it should be "Compared to women with lowest quintiles of LDL and total cholesterol"rather than "compared to women with ≥2 pregnancies".

Response: Thank you for noticing this. We have now changed the appropriate sections of the abstract according to your comments. (Please see page 2, line 47).

---

## [Editor Report · Decision Letter 2]

25 Sep 2019

Lipid levels after childbirth and association with number of children: a population-based cohort study

PONE-D-19-14622R2

Dear Dr. Pirnat,

We are pleased to inform you that your manuscript has been judged scientifically suitable for publication and will be formally accepted for publication once it complies with all outstanding technical requirements.

With kind regards,

C Mary Schooling

Academic Editor

PLOS ONE
---

## [Editor Report · Acceptance letter]

7 Oct 2019

PONE-D-19-14622R2 

Lipid levels after childbirth and association with number of children: a population-based cohort study 

Dear Dr. Pirnat:

I am pleased to inform you that your manuscript has been deemed suitable for publication in PLOS ONE. Congratulations! Your manuscript is now with our production department. 

With kind regards,

on behalf of

Dr. C Mary Schooling 

Academic Editor

PLOS ONE